# Dramatic Wound Closing Effect of a Single Application of an iBTA-Induced Autologous Biosheet on Severe Diabetic Foot Ulcers Involving the Heel Area

**DOI:** 10.3390/bioengineering11050462

**Published:** 2024-05-06

**Authors:** Ryuji Higashita, Yasuhide Nakayama, Manami Miyazaki, Yoko Yokawa, Ryosuke Iwai, Marina Funayama-Iwai

**Affiliations:** 1Department of Cardiovascular Surgery, Wound Care Center, Yokohama General Hospital, Yokohama 225-0025, Japan; manami.mzk@gmail.com; 2Biotube Co., Ltd., Osaka 565-0842, Japan; y.nakayama@biotube.co.jp; 3Department of Plastic Surgery, Yokohama General Hospital, Yokohama 225-0025, Japan; youko200076@gmail.com; 4Institute of Frontier Science and Technology, Okayama University of Science, Okayama 700-0005, Japan; iwai.ous@gmail.com (R.I.); m-iwai@ous.ac.jp (M.F.-I.)

**Keywords:** diabetic foot ulcer, tissue-engineered biosheet, in-body tissue architecture, wound repair

## Abstract

Introduction: Chronic wounds caused by diabetes or lower-extremity artery disease are intractable because the wound healing mechanism becomes ineffective due to the poor environment of the wound bed. Biosheets obtained using in-body tissue architecture (iBTA) are collagen-based membranous tissue created within the body and which autologously contain various growth factors and somatic stem cells including SSEA4-posituve cells. When applied to a wound, granulation formation can be promoted and epithelialization may even be achieved. Herein, we report our clinical treatment experience with seven cases of intractable diabetic foot ulcers. Cases: Seven patients, from 46 to 93 years old, had large foot ulcers including in the heel area, which were failing to heal with standard wound treatment. Methods: Two or four Biosheet-forming molds were embedded subcutaneously in the chest or abdomen, and after 3 to 6 weeks, the molds were removed. Biosheets that formed inside the mold were obtained and applied directly to the wound surface. Results: In all cases, there were no problems with the mold’s embedding and removal procedures, and Biosheets were formed without any infection or inflammation during the embedding period. The Biosheets were simply applied to the wounds, and in all cases they adhered within one week, did not fall off, and became integrated with the wound surface. Complete wound closure was achieved within 8 weeks in two cases and within 5 months in two cases. One patient was lost due to infective endocarditis from septic colitis. One case required lower leg amputation due to wound recurrence, and one case achieved wound reduction and wound healing in approximately 9 months. Conclusions: Biosheets obtained via iBTA promoted wound healing and were extremely useful for intractable diabetic foot ulcers involving the heel area.

## 1. Introduction

Diabetic foot ulcers are some of the most significant diabetic complications. Approximately 25% of people with diabetes will develop a lower-extremity ulcer over their lifetime [1]. Concomitant conditions associated with diabetes, such as peripheral artery disease causing limb ischemia, neuropathy causing sensory disturbance and foot deformity, and high blood glucose increasing the risk of infection, contribute to the causes of foot ulcers and gangrene. In addition, the slow healing nature of diabetic ulcers has been demonstrated [2], and the delayed healing of ulcers increases the need for amputation, which, in turn, increases morbidity and healthcare costs and simultaneously reduces an individual’s productivity and quality of life. Wound healing is an intricate and complex process. Chronic wounds caused by diabetes or lower-extremity artery disease are intractable, as the wound-healing mechanism becomes ineffective because of the poor environment of the wound bed. Moist dressings, debridement, wound offloading, and infection control are standard in the management of lower-extremity ulcers, yet even with the best standard care, these wounds are notoriously slow to heal, requiring many months of treatment [3]. The ideal therapeutic product would need to work for multiple targets in the wound repair process, such as by providing a matrix for cellular migration and proliferation, containing a number of essential growth factors and cytokines, and promoting increased healing and enhancement of the wound healing process. In addition, it would be ideal if it was non-immunogenic, reduced inflammation and scar tissue, provided a natural biological barrier, and demonstrated cost-effectiveness.

We developed an in vivo tissue-engineering technique called in-body tissue architecture (iBTA), enabling tissue preparation for autologous implantation by subcutaneously embedding a specifically designed mold [4,5,6,7]. A “Biosheet” can be obtained using iBTA, a collagen-based membranous tissue that is created within the body and autologously contains various growth factors, cytokines, and somatic stem cells, including SSEA4-positive cells. When applied to a wound, granulation formation can be promoted and epithelialization can even be achieved. Herein, we report our clinical treatment experiences with seven cases of intractable diabetic foot ulcers.

## 2. Cases

These procedures were approved by the ethics committee of Yokohama General Hospital (Approval Code: Yokorin-202249, Approval Date: 22 December 2022). The patients also provided written informed consent with regard to their case details and imaging studies.

All seven patients were diagnosed with type 2 diabetes mellitus, and four of these patients also had peripheral artery disease (Table 1). In addition, six were undergoing dialysis treatment for end-stage renal disease.

Revascularization and standard wound treatment had already been performed, and spinal cord stimulation was performed for Case 3, negative pressure wound therapy was performed for Case 4, and LDL apheresis was performed for Cases 4 and 5 as an additive wound treatment; however, foot ulcers are difficult to cure. 

Two or four Biosheet-forming molds for each patient were embedded subcutaneously in the chest or abdomen, and after 3 to 6 weeks, the molds were removed. Biosheets that formed inside the mold were retrieved and applied directly to the wound surface. Then, the Biosheets were covered with a nonadherent silicon dressing (Mepitel-One; Moneliche Health Care, Gothenburg, Sweden) or an acetylcellulose dressing (Sorbact: Century Medical, Inc. Tokyo, Japan), and were then overlapped with an antimicrobial non-woven hydrofiber fabric (Aquacel Ag Advantage: ConvaTec Inc., Oklahoma City, OK, USA).

Following the application of the Biosheets, dressings were exchanged weekly.

## 3. Results

In all cases, there were no problems with the mold embedding and removal techniques, and the Biosheets formed without any infection or inflammation during the embedding period. The Biosheets were simply applied to the wounds, and in all cases, they adhered within one week, did not fall off, and became integrated with the wound surface. Specific case examples are shown below.

Case 1 (Figure 1 and Figure 2): A 46-year-old man was obese with a height of 180 cm and a weight of 135 kg. He had had diabetes since his 20s, which was complicated by retinopathy, nephrotic syndrome, hypertension, and dyslipidemia. Both of his fifth toes had already been amputated because of diabetic ulcers. At the beginning of December 2022, an ulcer formed on the dorsum and heel of his left foot. By mid-December, the patient had developed a fever and redness and swelling of the entire left foot. Thus, he visited our wound care center on December 23.

A blood test showed a high inflammatory state (WBC was 15,700/mm^3^, CRP was 16.23 mg/dL). Antibiotic treatment was begun and then debridement was performed (Figure 1A). When the wound bed preparation was complete (Figure 1B(a)), two molds were embedded on 18 January 2023 (Figure 1C). The mold (length: 5 cm, diameter: 2 cm, supplied by Biotube Co., Tokyo, Japan) was a cylindrical shape made of porous stainless steel. Negative pressure wound therapy was performed during the mold-embedding period (Figure 1B(b)). Three weeks later, molds were removed, and Biosheets were attached to the wound (Figure 1C). As shown in Figure 2, Biosheets adapted in one week, and the wound had epithelialized after about 2 months.

Case 3 (Figure 3 and Figure 4): A 72-year-old man presented with chronic limb-threatening ischemia with gangrene of the left first toe and bilateral heel ulcers (Figure 3A). He had diabetes, end-stage renal disease, and high blood pressure and had already undergone right metatarsal amputation and coronary artery bypass surgery. In January 2023, the patient was referred for intractable bilateral heel ulcers and left first-toe gangrene. Lower-extremity angiography revealed occlusion of the right posterior tibial artery, 90% stenosis of the anterior tibial artery, occlusion of the left posterior tibial artery and peroneal artery, and 90% stenosis of the anterior tibial artery. His ischemic pain was also severe; therefore, spinal cord stimulation therapy was used to alleviate the pain and improve his peripheral microcirculation. After endovascularly treating the left lower limb, amputating the left thumb, and endovascularly treating the right lower limb, four molds were placed subcutaneously in the patient’s abdomen (Figure 3B). The molds (length: 6.4 cm, height: 1 cm, width: 2.5 cm, supplied by Biotube Co., Tokyo, Japan) had an elliptical cylindrical shape and were made of porous stainless steel. After 4 weeks, four Biosheets were applied to the wound (Figure 3C). The small heel ulcer healed in 3 weeks, and the large heel ulcer healed in 2 months (Figure 4).

Case 6 (Figure 5 and Figure 6): A 74-year-old man with chronic limb-threatening ischemia presented with gangrene of the right thumb and left forefoot. The patient had diabetes, end-stage renal disease, and hypertension; had a history of coronary artery disease, cerebral infarction, colon polyps, and thyroid cancer; and had undergone coronary artery intervention, open polypectomy, and thyroid cancer removal. During his previous hospitalization, he underwent lower-extremity endovascular treatment four times and was also given LDL apheresis treatment, but it was refractory. After admission to our hospital, antibiotics were administered, and left transverse total metatarsal amputation was performed to control the infection. Additional endovascular treatment was performed, and LDL apheresis was continued to improve the blood flow. However, the wound separated, and the metatarsal bones were exposed; therefore, we performed a left Chopard joint disarticulation, and at the same time, the right big toe was amputated and a stump was created. The right big toe wound healed, but the left stump wound dehisced again (Figure 5A). Four molds were embedded subcutaneously in the abdomen. The molds (length: 6.4 cm, height: 1.3 cm, width: 2.5 cm, supplied by Biotube Co., Tokyo, Japan) had an elliptical cylindrical shape made of porous stainless steel. Biosheets (Figure 5B) obtained by 6-week embedding of the molds were patched onto the left stump wound (Figure 5C). Wound granulation progressed, and healing was achieved after 5 months (Figure 6).

In summary, Biosheets in all cases showed quick adherence, within one week after application, and the wound surface proceeded to granulation tissue formation in two to three weeks and epithelization started simultaneously from the edge around the wound. There were no adverse events related to Biosheets during the follow-up period. Figure 7 shows that complete wound closure was achieved within 8 weeks in two cases (Cases 1 and 3) and within 5 months in two cases (Cases 4 and 6). Case 2 was lost due to infective endocarditis from septic colitis. Case 5 required lower leg amputation due to wound recurrence, and Case 7 achieved wound reduction and wound healing in approximately 9 months because of significant edema in her lower leg.

## 4. Discussion

It is difficult to treat ulcers and gangrene located on the heel, and it is recommended that primary amputation be carried out in select individuals with heel necrosis [8,9]. Soderstorm et al. [10] showed that ischemic tissue lesions located on the mid- and hindfoot had significantly prolonged ulcer-healing times. Kobayashi et al. [11] indicated that the healing rate of the heel wounds was lower than that of toe wounds and that they took a considerably longer time to heal. The wounds in all of our cases were located on the heel or mid- or hindfoot and seemed to be hard to heal; however, Biosheets were practically effective in promoting healing against the heel or mid- and hindfoot intractable wounds. 

Our previous study, using an animal experiment, showed that the tissue obtained by iBTA consisted of not only type I collagen fibers and fibroblast cells, but also type III collagen fibers and primitive somatic stem cells expressing CD90 and SSEA4 [12]. Another tissue analysis of Biosheets showed they included 560,000 CD90-positive mesenchymal stem cells/cm^2^ and 160,000 SSEA4-positive stem cells/cm^2^. In addition, component analysis showed that Biosheets presented many growth factors, such as platelet-derived growth factor, fibroblast growth factor, vascular endothelial growth factor, hepatocyte growth factor, and epidermal growth factor, which are thought to have considerable effects on wound healing. 

Advanced therapies such as bioengineered skin substitutes, xenografts, and allografts have been shown to promote wound closures, resulting in more consistent and faster healing of diabetic ulcers compared with standard therapies [13,14,15,16,17]. However, even though net cost savings can be achieved through increased healing rates, a faster time to healing, and reduced incidences of infection and amputation, these advanced therapies are expensive. Biosheets are much less expensive and safer because they are autologously produced and do not require factory instruments, cell culturing, or labor. In addition, wound management costs include supplies and dressings (15–20%), nursing time (30–35%), and hospitalization (more than 50%) [18]. Frequent dressing changes increase the cost of wound care; in comparison, Biosheets can be single-use and dressing changes may only be required about once a week. Even if there is a concern about Biosheets not being available off the shelf, or in other words, it taking some time to form an autologous tissue, their excellent cost performance can overcome that.

Basically, iBTA is a regenerative medicine technology that uses one’s own body as an incubator. Since the tissue is autologously created, there is no immunoreaction, and the risk of cancer and infection is thought to be low. Biosheets, therefore, should be safe.

## 5. Limitations and Future

We do not known about the mechanisms of the effectiveness of Biosheets on wound healing yet. Based on these experiences, we initiated a single-arm, exploratory, investigator-initiated clinical trial titled “To evaluate the safety and efficacy of the subcutaneous implantation of pluripotent stem cell accumulators (BSM1) and patch treatment of diabetic foot ulcers using Biosheets obtained from BSM1” in place since September 2023. The aims of this trial are to find the most effective treatment methods, such as setting the embedding period and the embedding site of the mold, and defining the optimal wounds. Simultaneously, we have analyzed the components of the Biosheet, elucidated the active ingredients, and clarified the healing mechanism. Our findings should be confirmed and expanded with a subsequent prospective randomized comparative trial.

## 6. Conclusions

Biosheets obtained via iBTA have potential to be the ideal wound repair products. In our cases, the results demonstrated their effectiveness in promoting wound healing and showed that Biosheets could be extremely useful for intractable diabetic foot ulcers involving the heel area.

## Figures and Tables

**Figure 1 bioengineering-11-00462-f001:**
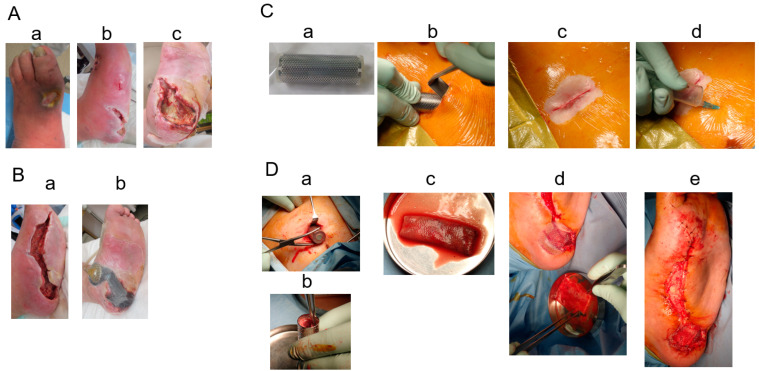
(**A**): Left foot ulcers at admission: **a**. ulcer on dorsal pedis; **b**. medial part; **c**. heel ulcer on left foot. (**B**): **a**. Wounds after debridement; **b.** negative-pressure wound treatment. (**C**): Subcutaneous mold-embedding procedure—**a**. the mold; **b**. subcutaneous embedding of the mold; **c**. closed wound over embedding; **d**. air removal of the inside mold through a needle puncture. (**D**): Biosheet patch therapy—**a**. removal of the mold from the subcutaneous space; **b**. obtained Biosheet; **c**. patch therapy using the opened Biosheet; **d**. wound covered by a siliconsheet (Mepitel One); **e**. finishing of patch therapy.

**Figure 2 bioengineering-11-00462-f002:**
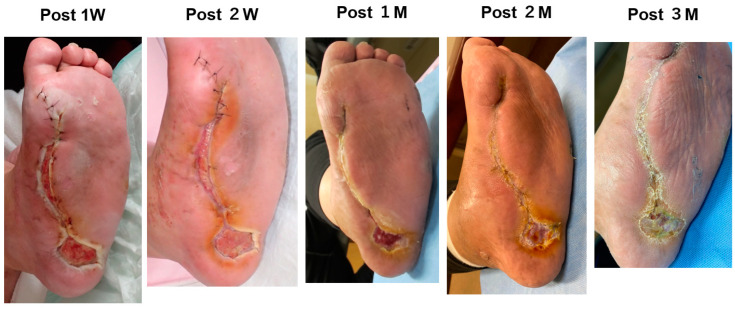
Post-operative course in Case 1. The wound healed in three months.

**Figure 3 bioengineering-11-00462-f003:**
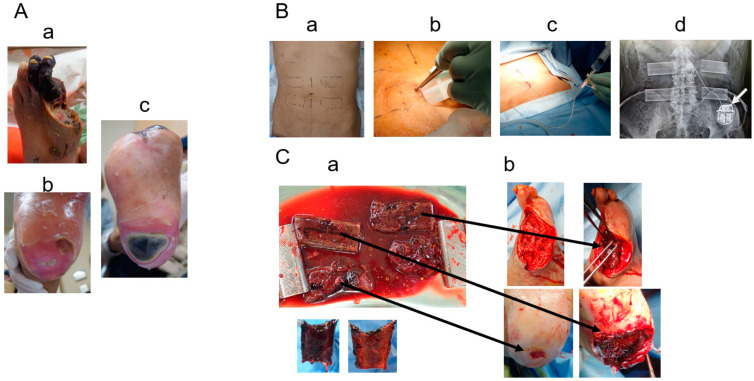
(**A**): Foot ulcers and gangrene in Case 3 upon admission—**a**. left 2^nd^- and 3^rd^-toe gangrene; **b**. left-heel ulcer; **c**. right-heel ulcer. (**B**): Subcutaneous mold-embedding procedure—**a**. the plan of the mold embedding; **b**. embedding procedure using a sizer; **c**. air removal from the subcutaneous space and inside the mold through a drainage tube after embedding; **d**. X-ray photograph after embedding the four molds. The white arrow indicates the spinal code stimulation generator. (**C**): Biosheet patch therapy—**a**. four Biosheets obtained after removing the mold from the subcutaneous space; **b**. Biosheets applied to the debrided wounds.

**Figure 4 bioengineering-11-00462-f004:**
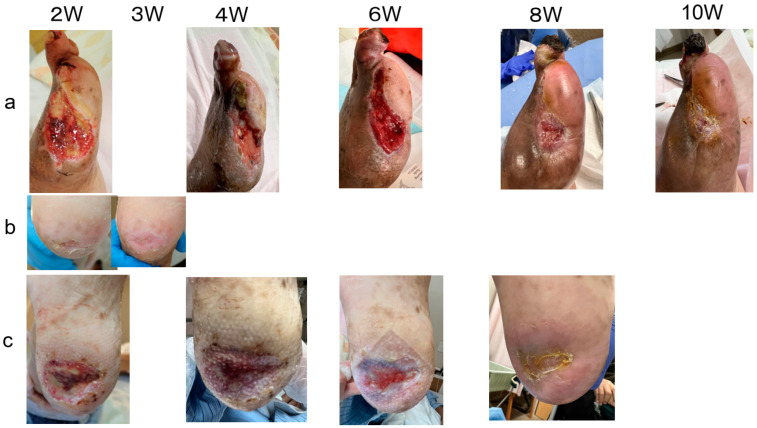
Post-operative course in Case 3. (**a**). The wound healed in ten weeks after the toes were debrided. (**b**). Small wound on the left heel healed in three weeks. (**c**). Large wound on the right heel healed in eight weeks.

**Figure 5 bioengineering-11-00462-f005:**
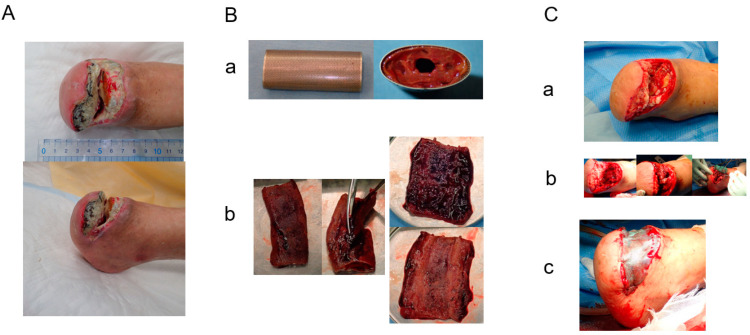
(**A**): Opened wound after Chopart amputation in Case 6. (**B**): **a**. Removed mold; **b**. Biosheet cut and opened. Upper right shows the inner surface and lower right shows the outer surface. (**C**): **a**. Debrided wound; **b**. three Biosheets applied; **c**. wound covered with an antimicrobial acetylcellulose dressing (Sorbact).

**Figure 6 bioengineering-11-00462-f006:**
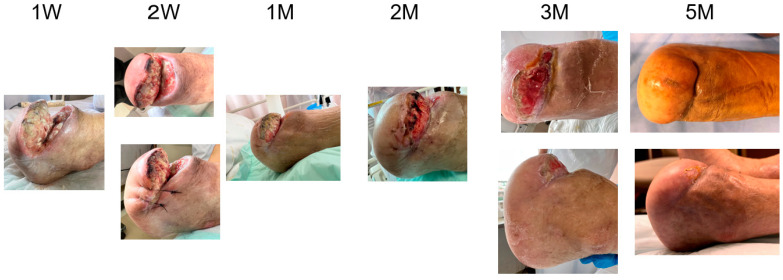
Post-operative course in Case 6. Wound closed in five months.

**Figure 7 bioengineering-11-00462-f007:**
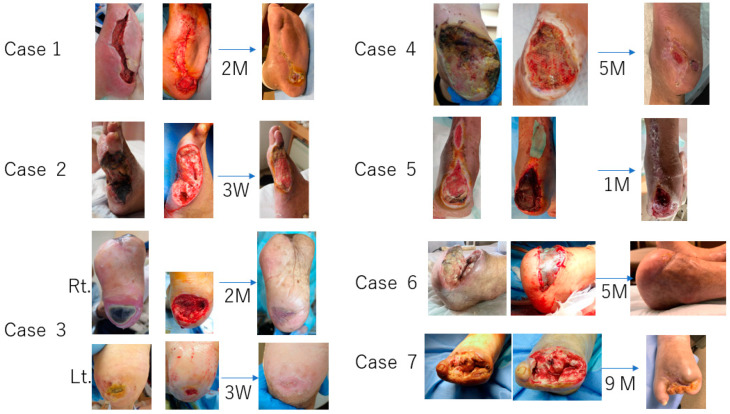
Summary of 7 Cases.

**Table 1 bioengineering-11-00462-t001:** Profiles and results of the patients.

Case	Age	Gender	Responsible Disease	Significant Medical Conditions	Revascularization	Additive Wound Treatment	Results (Dulation)
1	46	M	Diabetic neuropathy	DM, Obesity, HTN, DL		NPWT	healed (2M)
2	93	M	Diabetic microvasculopathy	DM, CKDG5D, CI, HTN, AF			death with IE
3	73	M	CLTI (PAD)	DM, CKDG5D, CAD, HTN	EVT	SCS	healed (2M)
4	53	M	CLTI (PAD)	DM, CKDG5D, CAD, CI	EVT	NPWT	healed (5M)
5	57	M	Diabetic neuropathy	DM, CKDG5D, CAD		LDL apheresis	BKA
6	74	M	CLTI (PAD)	DM, CKDG5D, CAD, AS(AVR), CI, HTN	EVT	LDL apheresis, PRP	healed(5M)
7	69	F	CLTI (PAD)	DM, CKDG5D	OS		healed (9M)

AF: atrial fibrillation, AS: aortic stenosis, AVR: aortic valve replacemnt, BKA: below-knee amputation, CAD: coronary artery disease, CI: celebral infarction, CKDG5D: chronic kidney disease group 5D, CLTI: chronic limb threatening ischemia, DM: diabetis melitus, DL: dislipidemia, EVT: endovascular treatment, HTN: hypertention, IE: infective endocarditis, LDL: low density lipoprotein, NPWT: negative pressure wound therapy, OS: open bypass surgery, PAD: peripheral artery disease, PRP: platelete rich plasma, SCS: spinal code stimulation.

## Data Availability

The data are not publicly available due to their containing information that could compromise the privacy of research participants.

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
