# Peer review of "Dramatic Wound Closing Effect of a Single Application of an iBTA-Induced Autologous Biosheet on Severe Diabetic Foot Ulcers Involving the Heel Area"

_bioengineering, 2024, doi:10.3390/bioengineering11050462_

Round 1

Reviewer 1 Report

Comments and Suggestions for Authors

What is the key point of this work?

What is the logical application of iBTA in diabetic foot ulcers and what is the advantage of the present application over the other matetials?

Comments on the Quality of English Language

there is no comments for language.

Reviewer 2 Report

Comments and Suggestions for Authors

Dear Authors,

Congratulations on your paper, and thank you for giving me the opportunity to review your work. Please consider my suggestions to enhance your manuscript.

The employment of in-body tissue architecture (iBTA) to produce autologous collagen-based membranous tissues (Biosheets), enriched with growth factors and stem cells, presents a highly innovative and pioneering approach. The case series design with seven patients provides valuable preliminary data but lacks the statistical power to draw generalized conclusions

1. The manuscript contains instances of language use that should be reassessed, along with specific dates and minor errors needing revision (e.g., "202X").

2. Results: It reports on the successful closure of wounds, a more detailed analysis of the healing process, including granulation tissue formation, time to epithelialization, and any adverse events, would provide a deeper understanding of the treatment

3. The discussion is poor should be improved:

A more explicit acknowledgment of the study's limitations, would provide a balanced view and underscore the need for further research.

It's needed a more detailed comparison with standard of care and specific advanced wound care technologies (e.g., bioengineered skin substitutes, xenografts) could provide a clearer picture of where this new method stands in the spectrum of available treatments.

4. Conclusion needs to be improved:

Although the study's results are promising, the conclusion could be strengthened by explicitly calling for further research.

Write about the need more extensive studies are needed to validate and expand upon the results.

The conclusion does not mention the study's limitations, which could give readers an incomplete view of the research. 

Would be important to write about the effectiveness, but should enphatize safety.
